# Polyphenol-Enriched Blueberry Preparation Controls Breast Cancer Stem Cells by Targeting FOXO1 and miR-145

**DOI:** 10.3390/molecules26144330

**Published:** 2021-07-17

**Authors:** Jean-François Mallet, Roghayeh Shahbazi, Nawal Alsadi, Chantal Matar

**Affiliations:** 1Cellular and Molecular Medicine Department, Faculty of Medicine, University of Ottawa, Ottawa, ON K1H 8M5, Canada; jmallet@uottawa.ca (J.-F.M.); rshah017@uottawa.ca (R.S.); nalsa068@uottawa.ca (N.A.); 2School of Nutrition, Faculty of Health Sciences, University of Ottawa, Ottawa, ON K1H8M5, Canada

**Keywords:** polyphenols, breast cancer stem cells, epigenetics, microRNAs, FOXO1, N-RAS

## Abstract

Scientific evidence supports the early deregulation of epigenetic profiles during breast carcinogenesis. Research shows that cellular transformation, carcinogenesis, and stemness maintenance are regulated by epigenetic-specific changes that involve microRNAs (miRNAs). Dietary bioactive compounds such as blueberry polyphenols may modulate susceptibility to breast cancer by the modulation of CSC survival and self-renewal pathways through the epigenetic mechanism, including the regulation of miRNA expression. Therefore, the current study aimed to assay the effect of polyphenol enriched blueberry preparation (PEBP) or non-fermented blueberry juice (NBJ) on the modulation of miRNA signature and the target proteins associated with different clinical-pathological characteristics of breast cancer such as stemness, invasion, and chemoresistance using breast cancer cell lines. To this end, 4T1 and MB-MDM-231 cell lines were exposed to NBJ or PEBP for 24 h. miRNA profiling was performed in breast cancer cell cultures, and RT-qPCR was undertaken to assay the expression of target miRNA. The expression of target proteins was examined by Western blotting. Profiling of miRNA revealed that several miRNAs associated with different clinical-pathological characteristics were differentially expressed in cells treated with PEBP. The validation study showed significant downregulation of oncogenic miR-210 expression in both 4T1 and MDA-MB-231 cells exposed to PEBP. In addition, expression of tumor suppressor miR-145 was significantly increased in both cell lines treated with PEBP. Western blot analysis showed a significant increase in the relative expression of FOXO1 in 4T1 and MDA-MB-231 cells exposed to PEBP and in MDA-MB-231 cells exposed to NBJ. Furthermore, a significant decrease was observed in the relative expression of N-RAS in 4T1 and MDA-MB-231 cells exposed to PEBP and in MDA-MB-231 cells exposed to NBJ. Our data indicate a potential chemoprevention role of PEBP through the modulation of miRNA expression, particularly miR-210 and miR-145, and protection against breast cancer development and progression. Thus, PEBP may represent a source for novel chemopreventative agents against breast cancer.

## 1. Introduction

Breast cancer is the world’s most commonly diagnosed cancer, making up 11.7% of total cases in 2020. Reproductive and hormonal factors, as well as lifestyle risk factors such as alcohol intake, obesity, and physical inactivity, contribute to breast cancer pathogenesis [1]. Histological and molecular subtypes are used to classify breast cancer. Molecular subtype classification is based on the presence or lack of estrogen receptors, progesterone receptors, and human epidermal growth factor receptor-2 (HER2) [2]. Triple-Negative is an aggressive subtype of breast cancer and because of the absence of estrogen, progesterone, and HER2 receptor expression, no targeted therapy has been developed for this subtype therapy [3].

In recent years, different targeted therapies, such as targeting epigenetic changes associated with cancer, have been developed to increase the efficacy of cancer treatment [2]. Epigenetic mechanisms appear to play a key role in cancer establishment and progression, and their deregulation has been reported at multiple levels, including DNA methylation, histone modifications, and, indirectly, microRNAs (miRNAs) expression [4,5,6]. In contrast to genetic mechanisms, epigenetic changes are reversible and are therefore capable of being targeted for intervention and cancer therapy, and can therefore be used as a biomarker for early detection of cancer [2,7,8].

Mounting evidence supports the idea that healthy eating habits can control and reduce many epithelial cancers, including breast cancer [9,10]. In this sense, natural products such as probiotics, polyphenols, and fermented plant foods are known for their anti-inflammatory effects [11,12,13] and may control neoplasia [14]. Blueberries are a well-known source of antioxidants and polyphenols [15]. Polyphenols found in blueberries are shown to reduce inflammation, oxidative stress [12], and metastasis [16], while promoting apoptosis in cancer cells [17].

During the fermentation process, bioefficacy, bioavailability, and content of blueberry polyphenols increase [12]. We have previously shown that fermentation of blueberry juice with *Rouxiella badensis* subsp. acadiensis (previously identified as *Serratia vaccinia*), a probiotic isolated in our lab from the natural microflora of lowbush blueberry, significantly raised the quantity of polyphenols present naturally in the juice [18]. This novel product, polyphenol enriched blueberry preparation (PEBP), was shown to reduce weight gain in the diabetic and obesity model KKA(y) mouse and exerted an antidiabetic effect by mimicking metformin anti-inflammatory effects. The effect was thought to be modulated by increasing AMPK activity and adiponectin level [19]. PEBP was also shown to prevent oxidative stress from hydrogen peroxide on neurons and to reduce nitric oxide production by macrophages [20,21]. Accordingly, we demonstrated that PEBP decreased the formation of cancer stem cells (CSCs) by controlling phosphatase and tensin homolog/phosphatidylinositol-3 kinase/protein kinase B (PTEN/PI3K/AKT), interleukin 6/ signal transducer, and activator of transcription 3 (IL-6/STAT3), and mitogen-activated protein kinase (MAPK) pathways, which are central nodes in CSC signaling and homeostasis [22].

Various mechanisms have been attributed to the protective effects of naturally occurring compounds such as PEBP against breast cancer. Modulation of the CSC self-renewal pathways is one important mechanism [22]. CSCs are a subpopulation of cells within tumors that are highly tumorigenic and can form spheres, termed mammospheres, with a CD44+/CD24−/low phenotype, under defined culture conditions. These cells can self-renew, differentiate into various types of cells composing the tumor [23], and are believed to be a major cause of relapse in many cancers, having been found in a large number of cancers [24].

Epigenetic-specific changes in CSCs have previously been reported [25]. We have shown that cellular transformation, carcinogenesis, and stemness maintenance are regulated by epigenetic-specific changes that involve miRNAs [26]. miRNAs are small nucleotide sequences that influence gene translation and have emerged as critical regulators of CSCs in drug resistance and cancer metastasis [27]. They work by binding to a complementary sequence in the target mRNAs, either by blocking the ribosome from translating the mRNA or by cleaving the RNA with the help of the miRNAs ribonucleoprotein complex. Their expression changes in many malignancies [28], and some of them can function as tumor suppressors or oncogenes [29]. miRNA networks have been reported to create a permanent feedback loop involving nuclear factor-κB (NF-κB), let-7 miRNAs, IL-6, and STAT3, which induce and maintain the CSC state [30].

Given the significant role of epigenetic regulation in breast cancer formation and, as we have previously shown in the repression of breast CSCs by PEBP [22], the current study aimed to investigate the possible mechanism of action of PEBP against the development of breast cancer through the regulation of the miRNAs expression signature involved in cell proliferation, survival, and CSC self-renewal pathways in vitro using breast cancer cell lines.

## 2. Results

### 2.1. Effect of PEBP on miRNAs Exoression in 4T1 and MDA-MB-231 Cell Cultures

For the microarrays experiment, we first performed an MTT assay using different doses of PEBP. The dose–response curve showed a significant effect of PEBP ranging from 40 to 200 µM gallic acid equivalent (GAE) (data not shown). Then, 60 µM GAE was used to assay the miRNAs profile. We examined the expression levels of many miRNAs by microarrays in 4T1 cells exposed to 60 µM GAE of PEBP for 24 h. Our results revealed that several miRNAs associated with different clinical-pathological characteristics of breast cancer, such as stemness, invasion, and chemoresistance, were differentially expressed (Table 1). In particular, miR-210, the most consistently and robustly induced miRNA under hypoxia, which is generally over-expressed in solid tumors [31], was found to be highly downregulated. Furthermore, miR-145, a PI3K/AKT-cancer-associated miRNA [32], was over-expressed following exposure of the cells to the PEBP. miR-145 is associated with IL-6/STAT3 pathways and is under-expressed in breast cancer with high metastatic capability [33].

We selected miR-210 and miR-145 as two of the predominant miRNAs regulated by PEBP treatment which are involved in the PI3K/AKT and STAT3 signaling pathways, for our functional analysis. 4T1 and MDA-MB-231 cells were treated with 60 µM GAE of either PEBP or NBJ for 24 h. The validation study by qRT-PCR revealed and confirmed the downregulation of oncogenic miR-210 and upregulation of tumor suppressor miR-145 (Figure 1 and Figure 2). miR-210 was significantly down-regulated in both 4T1 and MDA-MB-231 cells exposed to 60μM GAE of PEBP (*p* < 0.001 and *p* < 0.01, respectively). This effect was not seen in 4T1 cells when exposed to NBJ (*p* > 0.05) but was reduced by 50% in MDA-MB-231 cell culture (*p* < 0.05) (Figure 1A and Figure 2A). Furthermore, miR-145 expression was significantly increased in both cell lines treated with PEBP (*p* < 0.05), while treating cells with NBJ did not significantly change miR-145 expression in cells (Figure 1B and Figure 2B).

### 2.2. Effect of PEBP on FOXO1 Expression in 4T1 and MDA-MB-231 Cell Cultures

For assaying the expression of target proteins, we first treated cell lines with different doses of NBJ and PEBP, ranging from 40 µM to 200 µM GAE to optimize the doses for Western blot analyzing (data not shown). Subsequently, 100 µM and/or 150 µM GAE were used in related experiments. To determine the level of forkhead box protein O1 (FOXO1), 4T1 and MDA-MB-231 cells were exposed to 100 µM and 150 µM GAE of NBJ or PEPB for 24 h. FOXO1 is a transcription factor of PI3K/AKT known to influence the expression of miR-145 [34]. PEBP has been shown to inhibit PI3K/AKT activation in three different breast cancer cell lines, potentially affecting miR-145 through FOXO1 [22]. FOXO1 was over-expressed in both cell lines exposed to 100 µM GAE (*p* < 0.001 for 4T1 and *p* < 0.01 for MDA-MB-231) or 150 µM GAE of PEBP (*p* < 0.001) (Figure 3A,B). While treatment with NBJ had no significant effect on FOXO1 expression in 4T1 cells (Figure 3B), exposure to 150 µM GAE of NBJ may significantly increase expression of FOXO1 in MDA-MB-231 cells (*p* < 0.05) (Figure 3B).

### 2.3. Effect of PEBP on N-RAS Expression in 4T1 and MDA-MB-231 Cell Cultures

N-RAS has been found to overexpress in some subtypes of breast cancer leading to the formation and progression of breast cancer [35]. To examine the effect of PEBP on N-RAS expression, 4T1 and MDA-MB-231 cells were treated with 100 µM GAE of NBJ or PEBP for 24 h. Expression of N-RAS was significantly reduced in 4T1 and MDA-MB-231 cells (*p* < 0.05) following 24 h treatment with PEBP (*p* < 0.05) (Figure 4A,B). In addition, a significant decrease was observed in the N-RAS expression in MDA-MB-231 cells treated with NBJ (*p* < 0.01), while N-RAS expression significantly raised in 4T1 cells in the presence of NBJ (*p* < 0.01) (Figure 4A,B).

### 2.4. Effect of miR-145 on N-RAS Expression in 4T1 and MDA-MB-231 Cell Cultures

miR-145 exhibits significant inhibitory activity against breast cancer malignancy and tumor growth through negatively regulating N-RAS signaling [33]. To examine the role of miR-145 in N-RAS expression, 4T1 and MDA-MB-231 cells were transfected with either a miR-145 mimic or a miR-145 inhibitor. Expression of N-RAS decreased in the presence of the miR-145 mimic and increased in the presence of the miR-145 inhibitor in both 4T1 and MDA-MD-231 cell lines, although the results were not significant compared with control (*p* > 0.05). There was a significant difference in N-RAS expression between groups transfected with the miR-145 inhibitor and the miR-145 mimic in both 4T1 and MDA-MB-231 cell cultures (*p* < 0.001 and *p* < 0.05, respectively) (Figure 5A,B).

## 3. Discussion

Although breast cancer is considered a complex disease with a multifactorial etiology, emerging data about the importance of diet in the prevention of breast cancer are currently the subject of intense research [36]. Adhering to a healthy eating style may be associated with a significant reduction in the risk of breast cancer [37]. Biological and epidemiological evidence supports an inverse association of polyphenols intake and breast cancer, with more emphasis on subclasses of individual compounds of phenolic acids and the risk of postmenopausal breast cancer [38,39].

Evidence indicates an early deregulation of epigenetic profiles during breast carcinogenesis [7,8,40]. In this sense, various dietary regimes are thought to modulate susceptibility to breast cancer by altering normal epigenetic states and reversing abnormal gene activation or silencing [41]. Our pioneering research in integrative oncology has shown that probiotic and prebiotic intake has tremendous immunoprotective and chemopreventative effects against breast cancer [42,43,44,45,46,47,48,49,50]. The underpinning mechanisms are thought to involve miRNAs and epigenetic-specific changes controlling breast cancer stem cells [25] and metastasis in vivo [22,26,51].

A panel of 38 miRNAs has been found to be differentially expressed between molecular subtypes of breast cancer [52]. Among aberrantly expressed miRNAs, miR-125 and miR-145 were significantly down-regulated, whereas miR-21 and miR-155 were up-regulated [53,54]. Recent studies have shown that natural agents, including resveratrol, could alter miRNA expression profiles [55], leading to the enhancement of the efficacy of conventional cancer therapeutics. In addition, it has been shown that the anti-metastatic effect of pomegranate juice on prostate cancer cells is partly due to the expression and up-regulation of anti-invasive miRNAs, such as miR-355, miR-205, and miR-200, whereas pro-invasive miRNAs such as miR-21 and miR-373 were down-regulated by the juice [56,57]. Furthermore, the anticancer effect of other natural compounds such as melatonin and tocotrienols by regulation of miRNAs expression, including miR-145 and miR-210, and miR-429 and their target genes has been reported in breast cancer cells. These genes are associated with apoptosis, cellular senescence, and cell proliferation [58,59].

Cytokine-mediated cross-talk, led by IL-6, has been reported to play a role in tumor-elicited inflammation and the development of CSCs [6,60]. More precisely, the IL-6 pathway is subject to epigenetic modifications involving STAT3 signal transduction [61]. The interface by which IL-6 controls stemness strongly involves miRNAs. An inverse relationship has been reported between let-7 miRNA and IL-6 expression in breast cancer tissues, suggesting the importance of inflammatory activation of miRNAs-related to IL-6 pathways and regulatory circuits in stemness [30,54,62].

In the current study, we performed miRNA profiling in CSC cultures of mammary carcinoma cell lines exposed to PEBP. We revealed that several miRNAs associated with different clinical-pathological characteristics of breast cancer, such as stemness, invasion and chemoresistance, were differentially expressed (Table 1). Some clusters of these regulated miRNAs, such as hypoxamirs (regulating hypoxia) and metastamirs (regulating metastasis), are strongly involved in sustaining the inflammatory microenvironment that resolves in neoplasia [63,64]. Importantly, we demonstrated that the most prominent hypoxamir, miR-210, was remarkably downregulated following treatment of breast cancer cell lines with PEBP. miR-210 is involved in hypoxia-induced aggressiveness and resistance of CSCs [65]. Importantly, the differential expression of this overlapping set of miRNAs reinforces the hypothesis that PEBP is controlling CSC development by the deactivation of STAT transcription factors which control pro-inflammatory cytokine production and aberrant oncogenic signaling pathways. Our data also indicated overexpression of let-7g, miR-195, and miR-145 tumor suppressors that inhibit invasion and metastasis [66,67,68]. We therefore postulate that PEBP induces epigenetic-specific changes by modulating miRNA regulatory networks (tumor-suppressive or oncogenic miRNAs) and inhibiting CSC-dependent survival/stemness pathways. We also identified miRNAs associated with IL-6/STAT3 pathways such as miR-365 and miR-145 in CSC cultures. Thus, as one of the predominant group of miRNAs regulated by PEBP treatment, miR-210 and miR-145, and related signaling, were selected for further functional analysis in this study. Validation studies by qRT-PCR revealed that miR-210 was substantially decreased in 4T1 and MDA-MB-231 cell cultures compared to the negative control, while miR-145 was significantly increased in MDA-MB-231 cell cultures compared to control.

miR-210 plays an important role in mammary tumorigenicity and is controlled by STAT3 transcriptional activity [30]. miR-210 is over-expressed in various human tumors and cancer cell lines in hypoxic conditions, a vital feature of the tumor microenvironment [69,70]. Hypoxia promotes genomic instability in tumor cells. miR-210 may likewise control the DNA repair capacity of tumor cells during hypoxia [71] by specifically decreasing pro-apoptotic signals [72]. In addition, hypoxia has been found to induce expression of vascular endothelial growth factor (VEGF), IL-6, and CSC signature genes such as Nanog and Oct4 with increased cell migration/invasion, concomitant with the upregulation of miR-210 expression in human pancreatic cancer cells [73]. Upregulation of miR-210 is associated with poor prognoses for breast cancer patients and plays a role in the cancer’s invasion and transition [74]. Furthermore, downregulation of miR-210 has been reported to significantly suppress cell viability, increase apoptosis rate, and enhance radiosensitivity in hypoxic human hepatoma and lung cancers [75,76].

Interestingly, miR-210 expression is correlated with metastasis of breast and melanoma tumors [77] under the control of STAT3 in mammary carcinoma [62]. This observation perfectly aligns with the results of our previous study since we reported that the probiotic-like product, fermented blueberry juice, decreased the formation of CSCs in different types of mammary carcinomas cell lines as well [22]. In a recent review, miR-210 was shown to be a part of five immune-related miRNAs that can subvert the physiological immune response toward oncogenesis [78]. Many identified targets of miR-210 such as suppressor anaphase-promoting complex, cyclin-dependent kinase 10, SERTA Domain Containing 2 are involved in cell cycle regulation and correlate with aggressiveness of breast cancer [79].

miR-145, a PI3K/AKT-cancer-associated miRNA, was also over-expressed in our study. miR-145 is under-expressed in breast cancer with high metastatic capability [33]. Since we have previously shown that the PEBP-inhibited PI3K/AKT pathway was also accompanied by a decrease in STAT3 activation [22], it could be argued that regulation of miR-145 will potentially lead to tumor control and regression. miR-145 is also regulated by Akt in a p53-dependent manner. Suppression of PI3K activity substantially increases p53 levels and, at the same time, induces miR-145 [80]. In particular, p53 is involved in the upregulation of the expression of tumor suppressor miRNAs such as let-7, miR-34, miR-145, miR-26, miR-30, and miR-146a [81].

FOXO1, a transcription factor generally known for its role in adipogenesis and the inhibition of glucose production in response to insulin, is a target of phosphorylation by AKT [82]. Since the PI3k/AKT axis is often constitutively activated in cancer [83] and its deactivator PTEN is often mutated or deleted in cancer, preventing it from repressing AKT signaling [84,85], FOXO1 is often repressed in cancer. FOXO1 is also essential in cell cycle regulation, stress resistance, and tumor suppression, all crucial in cancer stem cells [86]. In this study, FOXO1 was shown to increase in 4T1 and MDA-MB-231 cell cultures after treatment with PEBP compared to the control group in a dose-dependent manner. Lack of FOXO3A expression in breast cancer patients is associated with an increased recurrence rate [87]. Inactivation of FOXO3A by the PI3K/AKT pathway favors cell survival, proliferation, and expansion of the CSC population and increases self-renewal and tumorigenic capacity, such as enhanced mammosphere formation, inhibition of differentiation, and increase in CD133 expression [87]. FOXO1 also promotes the expression of miR-145 [34] to control multiple proteins associated with cancer.

N-RAS is a vital effector for tumor growth [88]. A correlation has been seen between high N-RAS levels and the most aggressive of breast cancer subtypes, the triple-negative phenotype [89]. In this study, we found that N-RAS was significantly down-regulated in 4T1 and MDA-MB 231 cell lines. The 4T1 cell represents a murine cell line mimicking the advanced stage of breast cancer, and MDA-MB-231 represents the triple-negative cell line. miR-145 has been reported to block the activation of AKT and ERK1/2 pathways, directly targeting N-RAS [33,68]. Figure 6 illustrates the possible mechanism of action of miR-145 and miR-210 in cancer cells.

In conclusion, our data validate the potential chemoprevention role of enriched polyphenol blueberry mixture through modulation of miRNAs, in particular miR-210 and miR-145. Our data indicate that these miRNAs may be involved in FOXO1 and NRAS modulation, breast cancer development, and progression. Therefore, polyphenol-enriched blueberry preparation may represent a novel complementary alternative medicine therapy.

## 4. Materials and Methods

### 4.1. Preparation of Blueberry Juices

Mature lowbush blueberries (Vaccinium angustifolium Ait.) were purchased from Cherryfield Foods Inc. (Cherryfield, Maine, USA) as fresh and untreated fruits. Blueberry juice was extracted by blending the fruit (100g) in a Braun Type 4259 food processor. The fruit mixture was then centrifuged at 500 × *g* for 10 min to remove fruit skin and insoluble particles. The resulting juice was sterilized using 0.22 µm Express Millipore filters (Millipore, Etobicoke, Ontario, Canada). *Rouxiella badensis* subsp acadiensis (*Canan SV-53*), (*Rouxiella badensis* subsp. acadiensis has been filed in a U.S. Provisional Application No. 62/916,921 entitled “Probiotics Composition and Methods” for its potential probiotic effects [90]) was cultured as previously described [18]. The juice was inoculated with a saturated culture of the bacterium corresponding to 2% of the total juice volume. After a four day fermentation period, the transformed juice was sterilized by 0.22 um filtration. The total phenolic content was then measured by the Folin–Ciocalteau method using gallic acid as standard and hence expressed as μM gallic acid equivalent (GAE). The total phenolic content was increased from 5.9 mM GAE to 30.7 mM GAE, confirming successful transformation. Blueberry and biotransformed blueberry juice have been partially characterized elsewhere [18,91].

### 4.2. Cell Culture

Murine 4T1, and human MDA-MB-231 cell lines were obtained from American Type Cell Collection (ATCC; Chicago, IL, USA). The cells were grown in RPMI-1640, media containing FBS (10%, *v*/*v*) (Sigma–Aldrich, Oakville, ON, Canada), penicillin/streptomycin (0.05 mg/mL) at 37oC in a humidified atmosphere with 5% CO_2_.

### 4.3. miRNA Profiling

The miRNA profiling experiment was completed in vitro in 4T1 breast cancer cells using the Affymetrix GeneChip miRNA array 2.0 and validated using real-time quantitative reverse transcription PCR.

### 4.4. Real-time Quantitative Reverse Transcription PCR

4T1 and MDA-MB-231 cell lines were treated with 60 µM GAE of either PEBP or non-fermented blueberry juice (NBJ) for 24 h. Then cells were collected, and samples’ RNA was extracted using miRNeasy kit (Qiagen, Toronto, ON, Canada). Samples underwent a reverse transcription reaction to produce cDNA using individual probes. The cDNA was synthesized by Moloney Murine Leukemia Virus (MMLV) reverse transcriptase (Invitrogen, Burlington, ON, Canada). The expressions of miR-145 and miR-210 were measured by RT-qPCR using Taqman primers (Applied Biosystems, Burlington, ON) and a FastStart Taq Polymerase (Roche, Mississauga, ON, Canada) in a CFX96 machine (Bio-Rad, Mississauga, ON, Canada). Gene expression was normalized to gene reference U6 small non-coding RNA (Applied Biosystems, Burlington, ON).

### 4.5. miRNAs Transfection

4T1 and MDA-MB-231 cells were cultured in a medium without any antibiotics until they achieved approximately 30% confluence. They were then transfected with a miR-145 mirVana™ mimic or inhibitor (Ambion, Burlington, ON, Canada) using Lipofectamine (Invitrogen, Burlington, ON, Canada). The media was changed after 17 h, and the cells were grown without Lipofectamine until they reached 80% confluence.

### 4.6. Western Blot Analysis

After treatment with PEBP or NBJ, cells were collected and lysed. Cell lysates were run on a 4–12% acrylamide gel, transferred to a PVDF membrane, and probed with anti-FOXO1, anti-N-RAS, and anti-β-tubulin (Cell Signaling Tech. Inc., Danvers, MA, USA). Bands were visualized via chemiluminescence using horseradish peroxidase-conjugated secondary antibodies (Jackson ImmunoResearch Laboratories, West Grove, PA, USA). Bands were quantified using β-tubulin as loading control using Bio-Rad Quantity One software.

### 4.7. Statistical Analysis

GraphPad Prism 5.0 software (GraphPad Software Inc., San Diego, CA, USA) was used to perform Statistical analysis. One-way analysis of variance (ANOVA) and Bonferroni’s post-hoc tests were used to compare groups. *p* ≤ 0.05 was considered statistically significant. Data are reported as mean ± SEM.

## Figures and Tables

**Figure 1 molecules-26-04330-f001:**
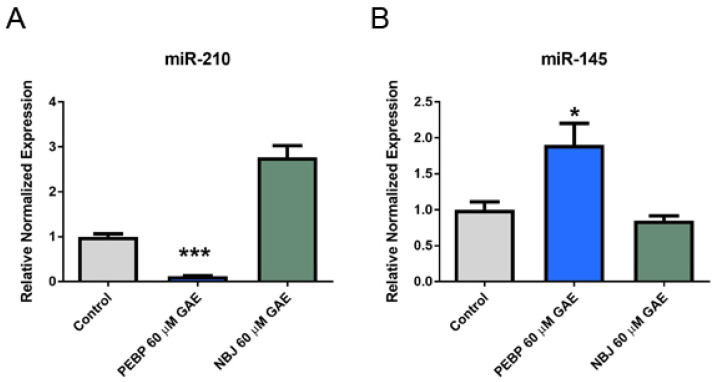
(**A**) Relative expression of miR-210 and (**B**) miR-145 by 4T1 cells after 24 h treatment with 60 µM gallic acid equivalent (GAE) of either polyphenol-enriched blueberry preparation (PEBP) or non-fermented blueberry juice (NBJ). The control consisted of using the same vehicle media used for the treatment groups. One-way ANOVA and Bonferroni’s post-hoc tests were used to compare groups. All values are mean ± SEM of 3 separate experiments. * *p* < 0.05 and *** *p* < 0.001 vs. control.

**Figure 2 molecules-26-04330-f002:**
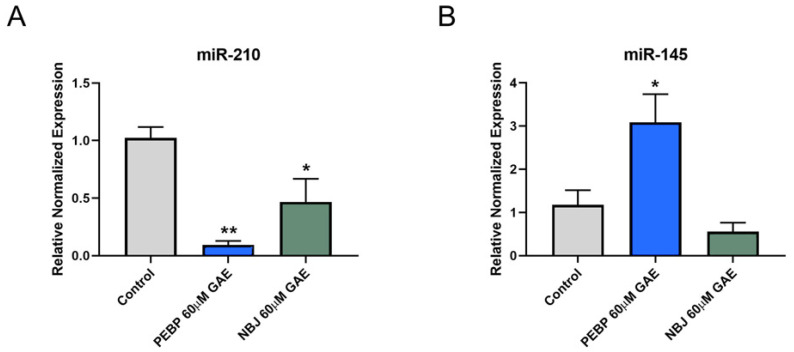
(**A**) Relative expression of miR-210 and (**B**) miR-145 by MDA-MB-231 cells after 24 h treatment with 60 µM gallic acid equivalent (GAE) of either polyphenol-enriched blueberry preparation (PEBP) or non-fermented blueberry juice (NBJ). The control consisted of using the same vehicle media used for the treatment groups. One-way ANOVA and Bonferroni’s post-hoc tests were used to compare groups. All values are mean ± SEM of 3 separate experiments. * *p* < 0.05 and ** *p* < 0.01 vs. control.

**Figure 3 molecules-26-04330-f003:**
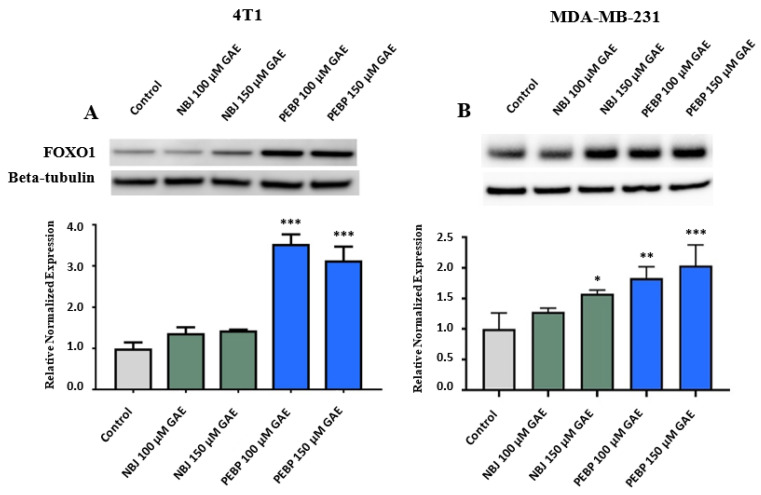
Relative expression of FOXO1 in (**A**) 4T1 and (**B**) MDA-MB-231 cells exposed to 100 µM or 150 µM gallic acid equivalent (GAE) of either polyphenol-enriched blueberry preparation (PEBP) or non-fermented blueberry juice (NBJ) for 24 h. The control consisted of using the same vehicle media used for treatment groups. Western blot images are from one representative experiment. One-way ANOVA and Bonferroni’s post-hoc tests were used to compare groups. All values are mean ± SEM of 3 separate experiments. * *p*< 0.05, ** *p* < 0.01 and *** *p* ≤ 0.001 vs. control.

**Figure 4 molecules-26-04330-f004:**
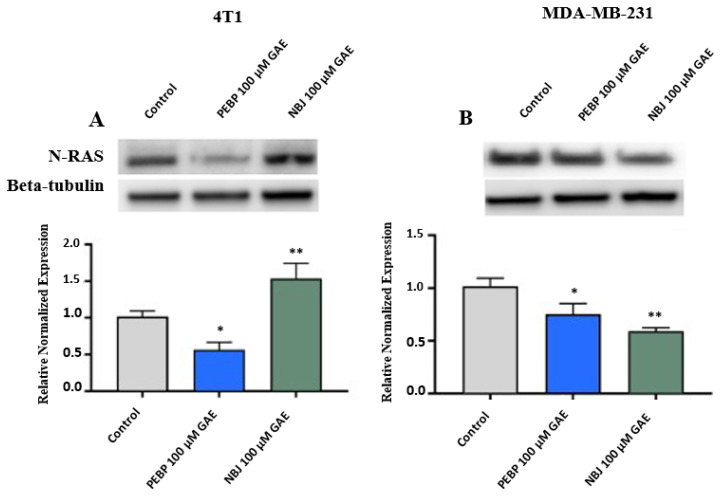
Relative expression of N-RAS in (**A**) 4T1 and (**B**) MDA-MB-231 cells exposed to 100 µM gallic acid equivalent (GAE) of either polyphenol-enriched blueberry preparation (PEBP) or non-fermented blueberry juice (NBJ) for 24 h. The control consisted of using the same vehicle media used for treatment groups. Western blot images are from one representative experiment. One-way ANOVA and Bonferroni’s post-hoc tests were used to compare groups. All values are mean ± SEM of 3 separate experiments. * *p* < 0.05 and ** *p* < 0.01 vs. control.

**Figure 5 molecules-26-04330-f005:**
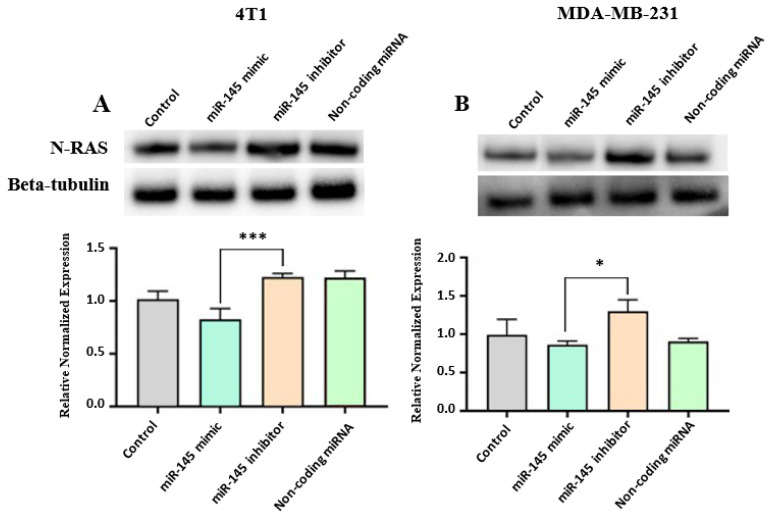
Relative expression of N-RAS in (**A**) 4T1 and (**B**) MDA-MB-231 cells transfected with a miR-145 mimic or inhibitor. The control consisted of using the same vehicle media used for treatment groups. Western blot images are from one representative experiment. One-way ANOVA and Bonferroni’s post-hoc tests were used to compare groups. All values are mean ± SEM of 3 separate experiments. * *p* < 0.05 and *** *p* < 0.001 miR-145 mimic vs. miR-145 inhibitor.

**Figure 6 molecules-26-04330-f006:**
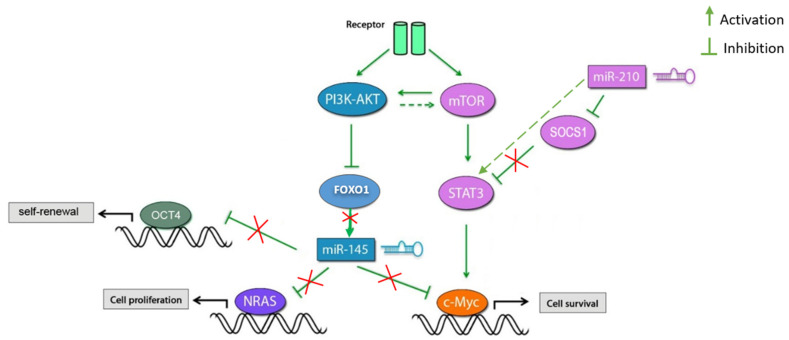
The potential mechanism of action of miR-145 and miR-210 in cancer cells. PI3k/AKT signaling is often constitutively activated in cancer. Inactivation of FOXO1 and subsequent inhibition of miR-145 expression by the PI3K/AKT pathway favors cell survival, proliferation, and expansion of the CSC population and increases self-renewal and tumorigenic capacity. Activation of STAT3 induces cancer cell survival by inducing the expression of c-Myc. miR-210 inhibits SOCS1 and activates the STAT3 pathway favoring cell survival.

**Table 1 molecules-26-04330-t001:** Expression of selected miRNAs in 4T1 cells exposed to 60 µM gallic acid equivalent (GAE) of PEBP for 24 h compared to non-treated cells.

Over-Expressed		Under-Expressed	
miRNAs	Fold Change	miRNAs	Fold Change
miR-145	3.04	miR-7	0.37
miR-34b	2.13	miR-450	0.40
miR-26a	1.97	miR-23b	0.44
miR-216b	1.95	miR-214	0.46
miR-101	1.86	miR-210	0.51
let-7g	1.77	miR-301	0.52
miR-150	1.73	miR-297	0.54
miR-365	1.70		
miR-195	1.65		
miR-182	1.44		

## Data Availability

The data presented in this study are available on request from the corresponding author.

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
