# Peer review of "Polyphenol-Enriched Blueberry Preparation Controls Breast Cancer Stem Cells by Targeting FOXO1 and miR-145"

_molecules, 2021, doi:10.3390/molecules26144330_

Round 1
Reviewer 1 Report
In the present manuscript, the authors aim to determinated the effect of Polyphenol enriched blueberry preparation (PEBP) on miRNAs expression and several target proteins. The manuscript is potentially interesting, however I have a few minor comments on presentation and two methodological concerns.
The authors tested the presence of miRNAs by microarray, and indicate that “We selected miR-210 and miR-145 as two of the predominant miRNAs regulated by PEBP”. It seems that there were several miRNA in both conditions: up and downregulated in cancer cells (Table 1). I am curious why the authors only investigate miRNA 210 and miRNA145 in these cancer cell lines. Other interesting candidates upregulated could be miR-34b and miR-301 or miR-297.
The authors then tested the effect of miRNA 210 and miRNA145 on FOXO1 and NRAS expression in 4T1 and MDA-MB231 cells exposed to 100 and 150 uM of PEBP. Please explain why used these concentrations while that the initial microarrays experiments were used 60uM.
Minor comments are the following:
The authors should include the labels in the WB figures (3-5) showing the treatment in each band.
Author Response
Special thanks to the reviewer because of the constructive comments and suggestions which help us to improve the quality of our work.
Please see the attachment for the response to the comments.

Reviewer 2 Report
The paper by Mallet and colleagues relies on the role of Polyphenol enriched Blueberry Preparation (PEBP) as a regulator of breast cancer (BC)-associated processes. Authors showed that PEBP differentially modulated miR-145 and miR-210 in two BC cell types and demonstrated that these miRNAs serve as direct targets for FOXO1 and N-RAS. They concluded that PEBP represents a source for novel chemopreventive agent against BC. Although this is an important manuscript, some adjustments are needed.
In Introduction section, more description on breast cancer presentation would be important, especially mentioning triple-negative BC which is the main intention of the study. Also, epigenetics approach as a tool for studying BC could be added in general.
Is there a dose-response curve (MTT assay) for testing better concentration of GAE? Why do authors choose for 60 µg for the microarray analysis and 100-150 µg for evaluating FOXO1? This is not clear for the readers.
How is the expression level of FOXO1 and N-RAS genes? This data could strengthen the Western blot results.
Figure 5. It is stated that both cell lines were exposed to “100 µM or 150 µM GAE (gallic acid equivalent)”. However, from the graphics, only exposured to 100 µM is shown. The concentrations that were used should be followed accordingly.
There are other important papers regarding differential expression of miRNAs 145 and 210 using other natural compounds on the breast cancer cells that might reinforce the discussion of the current data (see related papers PMID: 32910542, PMID: 30717416).
Minor comments
Define GAE when first mentioned in the text
Add the control vehicle used to treat control BC cells in figure legends. Also, add all statistical tests.
Standardize “Figure” instead of “figure”
Legend of “figure 5” is refered to “figure 1”.
Legend of Figure 6. Definition of activating and inhibiting processes should be added based on different arrows.
Author Response

(The authors gave the same response as above.)

Reviewer 3 Report
Please see the attached file

Author Response

(The authors gave the same response as above.)

Round 2
Reviewer 2 Report
No additional comments
Author Response
There are no additional comments to respond to.